



# Understanding Changes in Cloud Simulations from E3SM Version 1 to Version 2

Yuying Zhang[1], Shaocheng Xie[1], Yi Qin[2], Wuyin Lin[3], Jean-Christophe Golaz[1], Xue Zheng[1], Po-Lun Ma[2], Yun Qian[2], Qi Tang[1], Christopher R. Terai[1], and Meng Zhang[1]

[1]Lawrence Livermore National Laboratory, Livermore, CA, USA
[2]Pacific Northwest National Laboratory, Richland, WA, USA
[3]Brookhaven National Laboratory, Upton, NY, USA

*Correspondence to*: Yuying Zhang (zhang24@llnl.gov)

**Abstract.** This study documents clouds simulated by the Energy Exascale Earth System Model (E3SM) version 2 (E3SMv2)

and attempts to understand what causes the model behavior change in clouds relative to E3SMv1. This is done by analyzing

the last 30-year (1985-2014) data from the 165-year historical simulations using E3SMv1 and v2 and four sensitivity tests to

isolate the impact of changes in model parameter choices in its turbulence, shallow convection, and cloud macrophysics

parameterization (CLUBB), microphysical parameterization (MG2), and deep convection scheme (ZM), as well as model

physics changes in convective triggering. It is shown that E3SMv2 significantly improves the simulation of subtropical coastal

stratocumulus clouds (Sc) and clouds with optical depth larger than 3.6 over the stratocumulus to cumulus transition regimes,

where Shortwave Cloud Radiative Effect (SWCRE) is also improved, and the Southern Ocean (SO) while seeing an overall

slight degradation in low clouds over other tropical and subtropical oceans. The better performance in E3SMv1 over those

regions is partially due to error compensation between its simulated optically thin and intermediate low clouds for which

E3SMv2 actually improves simulation of optically intermediate low clouds. Sensitivity tests indicate that the changes in low

clouds are primarily due to the tuning made in CLUBB. The impact of the ZM tuning is mainly on optically intermediate and

thick high clouds, contributing to improved SWCRE and Longwave Cloud Radiative Effect (LWCRE). The impact of the

MG2 tuning and the new convective trigger is primarily on the high latitudes and the Southern Ocean (SO). They have a

relatively smaller impact on clouds than do the CLUBB and ZM tunings. This study offers additional insights about clouds

simulated in E3SMv2 by utilizing multiple data sets and the COSP diagnostic tool as well as through sensitivity tests. The

improved understanding will benefit the future E3SM developments.

## 1 Introduction

Given the importance of clouds in global radiative balance and hydrological cycle, continuously improving the representation

of clouds has been a key focus for global climate model (GCM) development. For instance, major efforts have been devoted

to reducing outstanding errors in clouds simulated in the U.S. Department of Energy (DOE)'s Energy Exascale Earth System

Model (E3SM) version 1 (E3SMv1) during development of its version 2 (E3SMv2) (Golaz et al. 2022). These efforts include





both improvements in representing its atmospheric physics and significant re-turning of atmospheric parameters in its deep convection, microphysics, and turbulence and macrophysics. As a result, the lack of stratocumulus (Sc) along the subtropical coasts, one outstanding error shown in E3SMv1 (Golaz et al. 2019, Rasch et al. 2019, Xie et al. 2018, Zhang et al. 2019), has been significantly reduced in E3SMv2, along with improvements in the stratocumulus to cumulus transition regimes. The updated physics and tuning parameters in E3SMv2 have also resulted in changes to other types of clouds over different cloud regimes such as the mixed-phase clouds in high latitudes (Zhang et al. 2022) and tropical high clouds (Golaz et al. 2022). Understanding model cloud behavior changes from E3SMv1 to E3SMv2 will provide necessary insights into clouds simulated by E3SMv2 and guide future E3SM development.

In this study, we perform a comprehensive evaluation of clouds simulated in E3SMv2. Our goal is to document the overall features in clouds simulated from this newly released model and understand what processes are primarily responsible for the changes from E3SMv1 to E3SMv2. To archive our goal, we conduct and investigate a series of sensitivity tests using its atmospheric model (EAM) to isolate the impact of changes made in atmospheric physics and parameter choices in E3SMv2. This study provides a more comprehensive picture on E3SMv2 performance on cloud simulations, which was beyond the scope of the E3SMv2 overview paper by Golaz et al. (2022).

The cloud evaluation presented in this study is performed using the community satellite simulator package - the Cloud Feedback Model Intercomparison Project (CFMIP) Observation Simulator Package (COSP; Bodas-Salcedo et al. 2011, Swales et al. 2018) to improve the consistency between model clouds and satellite observations. COSP contains several independent satellite simulators for better comparing model clouds with satellite measurements collected by the International Satellite Cloud Climatology Project (ISCCP; Rossow & Schiffer, 1999), the Moderate Resolution Imaging Spectroradiometer (MODIS), the Multiangle Imaging Spectro-Radiometer (MISR), Cloud-Aerosol Lidar and Infrared Pathfinder Satellite Observation (CALIPSO), and CloudSat. The use of satellite simulators will not only make a fairer comparison between model clouds and satellite data, but also allow a more in-depth analysis of clouds. For example, clouds can be assessed in terms of their optical properties and vertical location, which dictate their radiative effects.

The manuscript is organized as follows. Section 2 provides a brief description of E3SMv2 with emphasis on changes in its atmospheric physical parameterizations and parameters settings compared to E3SMv1. In addition, satellite data and sensitivity tests analyzed in the study are also discussed. Section 3 documents the overall features in clouds simulated in E3SMv2. The processes that primarily attribute to the changes in simulated clouds from E3SMv1 to E3SMv2 are discussed in Section 4. A summary is provided in Section 5.

## 2 Model, Data, and Sensitivity Tests

### 2.1 E3SMv2

E3SMv2 is the version 2 of E3SM that was released in 2021 for public use. Compared to its precedent version (E3SMv1), E3SMv2 shows improved computational efficiency (approximately twice as fast) and simulated climate specifically related to





clouds and precipitation (Golaz et al., 2022). The improved computational efficiency mainly results from the implementation of high-order, property-preserving semi-Lagrangian tracer transport (Bradley et al., 2021) and high-order, property-preserving

dynamics-physics-grid remap ("Physgrid") (Hannah et al., 2021). The improved simulation of clouds and precipitation is mainly attributed to the updates of atmospheric physics and parameter re-tuning. Here we emphasize the changes made in cloud and convection parameterizations in E3SMv2 since they are relevant to the improved simulation of clouds that are discussed in this study.

E3SMv2 uses the same set of atmospheric physics as E3SMv1 as described by Rasch et al. (2019) and Xie et al. (2018). Its

cloud and convection parameterizations include a third-order turbulence closure parameterization (CLUBB; Cloud Layers Unified By Binormals) (Golaz et al., 2002; Larson, 2017; Larson & Golaz, 2005) for representing processes related to planetary boundary layer turbulence, shallow convection, and cloud macrophysics. The deep convection scheme is based on Zhang & McFarlane (1995) (ZM hereafter) with a dilute convective available potential energy (CAPE) modification by Neale et al. (2008). An updated version of the Morrison and Gettelman (2008) scheme (MG2, Gettelman et al., 2015) is used for

representing cloud microphysics of stratiform and shallow convective clouds. The MG2 is combined with a classical nucleation theory-based ice nucleation (IN) parameterization for the heterogeneous ice formation in mixed phase clouds (Wang et al. 2014).

A notable update related to clouds and precipitation is the use of a new convective trigger function described by Xie et al. (2019) and Wang et al. (2020) in ZM to improve the simulation of precipitation and its diurnal cycle. The new convective

trigger named as dCAPE-ULL uses the dynamic CAPE (dCAPE) trigger developed by Xie and Zhang (2000) with an unrestricted air parcel launch level (ULL) approach used by Wang et al. (2015). It was designed to address the unrealistically strong coupling of convection to the surface heating in ZM that often results in unrealistically too active model convection during the day in summer season over lands and improve the model capability to capture mid-level convection for nocturnal precipitation. Other updates include the implementation of a convective gustiness scheme following Redelsperger et al. (2000)

to account for subgrid-scale surface wind gustiness and improve the representation of tropical clouds and precipitation (Harrop et al., 2018; Ma et al., 2022).

Significant model re-tunings have also been made to reduce errors in clouds and precipitation during the E3SMv2 development. Following Ma et al. (2022), several tuning parameters are recalibrated in CLUBB, ZM deep convection, and microphysics schemes to improve the simulation of cloud and precipitation. Information learned from the short Parameter Perturbation

Ensemble simulations (Qian et al. 2018) is also used to guide EAMv2 model re-tuning effort. See Table A1 in Golaz et al. (2022) for details about the tuning parameters used in E3SMv2 model and the difference from E3SMv1.

**2.2 Satellite data**

Satellite measurements from ISCCP, MODIS, MISR, and CALIPSO are used to evaluate clouds simulated by E3SMv2. CERES-EBAF Edition 4.1 data are used to evaluate model simulated TOA shortwave cloud radiative effect (SWCRE) and

longwave cloud radiative effect (LWCRE). More detailed information and references about these data are given in Table 1.



Zhang et al. (2019) discussed additional details about these measurements. Here we highlight some of the key points from Zhang et al. (2019) to facilitate interpreting the results of this study.

Measurements from the ISCCP, MODIS and MISR are from passive instruments. They contain the information about the area coverage of clouds stratified by cloud-top pressure (*ctp*) or cloud-top height (*cth*) of the highest cloud in a column and by the
column integrated cloud optical thickness ($\tau$), which can be summarized in joint histograms of *ctp*-$\tau$ or *cth*-$\tau$. As discussed in earlier studies (Marchand et al., 2010; Pincus et al., 2012), notable differences are found in the joint histograms among the three data sets, which are largely due to instrument limitations and different algorithms used for detecting clouds and retrieving the cloud heights and optical depths. For example, the ISCCP often has difficulties to detect small cumulus clouds and mistakenly put optically thin cirrus as a mid-topped cloud when there are low clouds underneath (Mace et al. 2006). In contrast,
the retrievals used in MODIS do not work well for low-level clouds under temperature inversions or broken low-level clouds (Pincus et al. 2012) although MODIS cloud data are usually considered as the most accurate for high-topped cloud among the three passive instruments. In addition, partly cloudy pixels are excluded in MODIS while they are treated as homogeneous and included in ISCCP detected cloud. Pincus et al. (2012) found that this could lead to 15% difference in optically thinnest clouds estimated by ISCCP and MODIS. Compared to ISCCP and MODIS, MISR gives the most accurate estimate of cloud top
height for low-level clouds and better detection of cumulus clouds. Different from ISCCP, MODIS, and MISR, CALIPSO uses active instruments to measure cloud height directly and therefore can provide information of cloud vertical structure. The CALIPSO data used in this study include high, middle, and low cloud fraction derived from the attenuated backscattered profile at 532 nm (Chepfer et al., 2010). Like Zhang et al. (2019), this study utilizes all these available observations to provide more complete information on model simulated clouds.

### 2.3 Sensitivity Tests

Table 2 lists the simulations analyzed in this study. In addition to the last 30-year (1985-2014) data from the 165-year historical simulations with E3SMv1 and E3SMv2 (Golaz et al. 2019 and 2022) used to document the overall features of clouds simulated in these models, we also analyze the sensitivity tests with the atmosphere model of E3SMv2 (EAMv2) to investigate the impact of each major change on the simulated clouds. The sensitivity tests are described by Qin et al. (2023). More details about what
parameters were tuned in v2 are given in the Table A1 of Golaz et al. (2022). All the sensitivity tests are 6-year AMIP-type runs with present-day (2010) forcing from the Intergovernmental Panel on Climate Change (IPCC) AR5 emission data set (Lamarque et al., 2010) along with climatological sea surface temperature and sea ice prescribed from the observations (repeating seasonal cycle without interannual variability). To accurately measure their impacts, the same 6-year AMIP runs were conducted for the default E3SMv2. The last 5 years from each run are analyzed. By comparing the AMIP runs with the
30-year coupled simulations, the short-term AMIP runs can well reproduce the major model errors shown in the long-term coupled simulations (not shown). This suggests that the E3SM cloud behaviors should be mostly controlled by its atmosphere model and most systematic errors in clouds are apparent in a few year simulations since clouds are associated with fast physics (e.g., Xie et al. 2012, Ma et al. 2014, and Qian et al., 2018).





## 3 General Cloud Features Simulated in E3SMv1 and E3SMv2

Results shown in this section are based on the last 30-year (1985-2014) data from the 150-year historical simulations, which are part of the E3SMv1 and E3SMv2 CMIP6 DECK plus historical simulation campaigns (Eyring et al., 2016). The exception is for MODIS simulator output. A bug was found in the MODIS simulator output shortly after the 150-year's historical simulations were completed. Therefore, the results from MODIS analyzed here are from the default E3SMv1 and E3SMv2 simulations conducted in the sensitivity tests, that is, the 6-year AMIP runs. As we will discuss later, the 6-year AMIP runs

can well represent the general features in cloud simulations as we see in the 30-year coupled runs. So, we do not expect this issue will affect what we will learn from this study. In this section, we will emphasize the overall features in clouds simulated in E3SMv2 by utilizing COSP. We focus on annual mean climatology of clouds simulated by both models and changes from E3SMv1 to E3SMv2.

### 3.1 Evaluation of model cloud with COSP

#### 3.1.1 Total cloud fraction

The annual mean total cloud fraction between the models and the ISCCP, MODIS, and CALIPSO observations is shown in Figure 1. Note that clouds with $\tau < 1.3$ in ISCCP and MODIS are neglected due to the large uncertainty of cloud detection from passive instruments as discussed earlier (Marchand et al. 2010; Pincus et al. 2012). The impact of the exclusion of the optically thin clouds in ISCCP and MODIS was discussed by Zhang et al. (2019). In general, the omission of the optically thin

clouds leads to a noticeable reduction of total cloud fraction between 60S and 60N, however, the exclusion of the optically thin clouds in MODIS has minor effect on its total cloud fraction (not shown). Beyond 60S and 60N, measurements from the passive instruments are less reliable due to their difficulties in detecting clouds over surfaces with ice and snow and therefore caution needs to be taken for model results discussed in these regions.

Overall, both models produce comparable results with slightly fewer clouds simulated in E3SMv2 compared to its previous

version. E3SMv2 generally shows slightly larger errors than E3SMv1 compared to ISCCP and CALIPSO. Relative to ISCCP, E3SMv2 underpredicts clouds in the tropical and extratropical regions and has more clouds over the Arctic than observations. As we will show later, this is mainly related to its simulated low clouds. E3SMv1 shows a similar error pattern in most regions except for the Pacific Ocean where clouds are generally overestimated. Relative to CALIPSO, both models underpredict clouds globally except for the Arctic. Compared to MODIS, however, E3SMv2 shows a slightly better result than E3SMv1 with

reduced biases over most regions. The discrepancy in model performance against different satellite data is likely not due to the use of 6-year AMIP simulations for MODIS since similar discrepancy is seen when 6-year simulation data are used for ISCCP and CALIPSO (not shown).

A robust improvement made in E3SMv2 is the considerable increase of stratocumulus cloud over the eastern ocean basins along the coasts in both hemispheres. The comparison with all three satellite observations shows this improvement. This is

significant since the lack of stratocumulus along the coasts in E3SMv1 (common in most climate models) is the one of the





main outstanding issues that the development of E3SMv2 tackled. Cloud biases over N. H. storm tracks, the stratocumulus to cumulus transition regimes, and the Southern Ocean (SO) are also noticeably reduced in E3SMv2 as seen in the comparisons particularly with ISCCP and MODIS.

The model simulated clouds are further examined by analyzing the column-integrated cloud optical depth distributions for
total cloud fraction averaged in the domain between 60S and 60N and over the SO region between 45S and 60S, respectively (Fig. 2). The domain between 60S and 60N is selected because measurements from ISCCP and MODIS are more reliable while the SO region is selected since most climate models have difficulties in accurately capturing clouds over this region where E3SMv2 shows clearly improvements in cloud simulations compared to E3SMv1. ISCCP and MODIS agree relatively better for clouds with $\tau > 3.6$, while for clouds with $\tau < 3.6$, the total cloud fraction in ISCCP is significantly larger than that in
MODIS, primarily due to the different assumptions on partly cloudy pixels in their retrievals (Zhang et al. 2019). For $3.6 < \tau < 23$, MODIS shows larger cloud fraction than ISCCP. In contrast, model results from the two simulators are very close to each other. This is because ISCCP and MODIS simulators use nearly the same method to determine the values of $\tau$ (Pincus et al. 2012).

Both models agree with MODIS much better than with ISCCP in both examined regions. Relative to MODIS, E3SMv1
generally well reproduces optically thin clouds ($1.3 < \tau < 3.6$) and notably overestimates optically intermediate to thick clouds ($\tau >= 9.4$). In contrast, E3SMv2 underpredicts the optically thin clouds and shows slightly larger overestimation of thick clouds ($\tau > 23$) compared to E3SMv1. The largest discrepancy between E3SMv1 and v2 is seen for $\tau$ between 3.6 and 9.4 where E3SMv2 performs worse than v1 and considerably underestimates the MODIS observed clouds. This suggests that some features seen in the total cloud fraction in Figure 1 are a result from compensating errors in clouds with different optical
properties. For example, the reduction of positive cloud fraction bias over the SO in E3SMv2 relative to MODIS, as shown in Figure 1, appears to stem more from the deficiencies in optically thin and intermediate clouds than the slightly more excessive optically thick clouds with respect to E3SMv1.

### 3.1.2 Cloud vertical structure

The annual mean high, middle, low cloud fraction from CALIPSO and model biases from the observation are shown in Figure
3. The high, middle, and low clouds in both models and CALIPSO are defined as cloud top pressure lower than 440mb, between 440mb and 680mb, and higher than 680mb, respectively. E3SMv1 and E3SMv2 show similar bias patterns for all types of clouds with clouds at all heights mostly underpredicted in the low latitudes and low clouds overpredicted in the northern high latitudes and the SO. Over the Tropics, E3SMv2 slightly improves high and middle clouds with noticeable reduction of errors in South Pacific (high clouds) and tropical and subtropical Pacific (middle clouds), when compared with
E3SMv1. On the other hand, it degrades the representation of low clouds over tropical and subtropical oceans except along the subtropical coasts in both Hemispheres where E3SMv2 produces much more stratocumulus (better). Over the SO, the overpredicted clouds in E3SMv1 are reduced in E3SMv2. The degradation of low clouds over tropical and subtropical oceans





is partially due to error compensation in low clouds with different optical properties in E3SMv1 as will be shown in a comparison with the MISR low cloud data, which give a more accurate estimate of low clouds over oceans compared to

CALIPSO (Zhang et al. 2019). Also over the SO E3SMv2 shows larger overestimation of middle-level clouds and smaller overestimation of low clouds. This breakdown of biases in terms of clouds vertical structure from CALIPSO provides additional information to understand the overall model performance in simulating total cloud fraction.

### 3.1.3 High clouds

The simulated high clouds in three optical thickness ranges (thin, intermediate, and thick) are further examined with MODIS

(Fig. 4) since MODIS provides more accurate information about high clouds than other datasets. Despite some reginal differences, E3SMv2 shows a very similar error pattern to E3SMv1. In general, both models overestimate the MODIS cloud fraction (particularly over land) for optically thin clouds and underestimate it (mainly over ocean) for optically intermediate and thick clouds. E3SMv2 shows slightly larger error for optically intermediate clouds and slightly smaller error for optically thick clouds. There are some improvements seen in E3SMv2 along the Antarctic coasts for optically intermediate clouds and

in the Arctic for optically thick clouds where E3SMv1 slightly overestimates the observed clouds.

### 3.1.4 Low clouds

As MISR can detect low clouds well, it is used to provide additional insights into model deficiencies in simulating low clouds. The optically thin, intermediate, and thick low cloud fractions (cth below 3 km) from MISR (only available over oceans) and the difference between the models and MISR are displayed in Figure 5. For optically thin low clouds ($0.3 < \tau < 3.6$), there is

little change in model errors from E3SMv1 to E3SMv2. Both models largely underestimate the optically thin low clouds in both tropical and midlatitude oceans. For optically intermediate low clouds ($3.6 < \tau < 23$), E3SMv2 dramatically improve the simulation. The large overestimation over the N.H. storm tracks, the SO, and the stratocumulus to cumulus transition regimes and underestimation in the Sc regions in E3SMv1 are significantly reduced in E3SMv2, consistent with the earlier discussions and Golaz et al. (2022). However, the smaller positive biases in optically intermediate low clouds in E3SMv2 leads to a bigger

negative bias in total low-cloud fraction than E3SMv1 as shown in Fig. 3 (compared to CALIPSO low clouds). The better performance shown in E3SMv1 is clearly due to error compensation between its simulated optically thin (underestimated) and intermediate (overestimate) low clouds. The optically thick low clouds are not the dominant cloud type. Only a few optically thick low clouds are detected by MISR, for which the models show a slightly positive bias in the N. H. storm tracks and the SO, especially for E3SMv2.

### 3.2 Cloud radiative effect


Clouds have a large impact on radiation. The biases in model simulated shortwave and longwave cloud radiative effect (SWCRE and LWCRE) from the CERES-EBAF Ed4.1 dataset (Fig. 6) are closely related to those in model clouds. SWCRE





is largely influenced by low clouds. The most noticeable improvement from E3SMv1 to E3SMv2 is over the stratocumulus regimes along the west coast of continents where the severely underestimated SWCRF in E3SMv1 is significantly reduced

due to the improvement of Sc as discussed earlier. In contrast, the biases in LWCRE are more related to high clouds. The major error in E3SMv1 is the much weaker LWCRE due to large underestimation of (optically intermediate) high clouds over the tropical Indo-West Pacific region (Figure 4e and 4f). This problem is reduced over the West Pacific region in E3SMv2 while slightly exaggerated over eastern Indian Ocean. For other regions, the change is little from E3SMv1 to E3SMv2.

## 4 Impact of major changes in v2 on cloud simulations

The previous section indicates that E3SMv2 largely improves the stratocumulus clouds over the eastern ocean basins in both hemispheres along the coast of the continents of South Africa and North and South America along with the improvement seen in the stratocumulus to cumulus transition regimes, and the Southern Ocean (SO) while it degrades the simulations of high clouds and total cloud fraction compared to E3SMv1. The improvements in low clouds are mainly from optically intermediate clouds with $\tau$ in a range of 3.6 and 23. For optically thin ($\tau < 0.3$) and thick ($\tau > 23$) low clouds, E3SMv2 actually produces

slightly worse result. The degradation of high clouds is primarily from the intermediate clouds.

In this section, we discuss what changes made in E3SMv2 have led to these changes in clouds from E3SMv1 to E3SMv2 through carefully designed sensitivity experiments, which are shown in Table 2 and discussed in Section 2.3. Since model responses to these changes are similar regardless of which simulator output is used, for simplicity, we use MODIS for total cloud fraction and high clouds, CALIPSO for vertical cloud structure, and MISR for low clouds. The inclusion of MODIS and

MISR clouds are particularly meaningful to aid in the breakdown of the model responses in terms of cloud opacity.

### 4.1 MODIS total cloud fraction

Figure 7 shows the impact of these changes on the simulation of total clouds. The improved stratocumulus along the west coast of continents (i.e., South Africa, North and South America) is clearly due to the re-tuning of CLUBB, which promote stratocumulus-like symmetric mixing by increasing the damping coefficients and allowing larger horizontal variation in

subgrid vertical velocity as described in Ma et al. (2022). However, the re-tuning of CLUBB also leads to a considerable reduction of clouds over tropical and subtropical oceans, which contributes to the fewer clouds produced in E3SMv2 compared to v1. This is mainly from tuning those parameters that can decrease boundary layer mixing and decoupling between boundary layer and free troposphere. The cloud lateral entrainment is also decreased in v2 due to the CLUBB-re-tuning, which could lead to a reduced cloudiness in shallow cumulus regime. The reduction of clouds is partially offset by the tuning made in ZM,

which extends clouds originated from deep convections to almost everywhere. This is largely due to the tuning of the autoconversion for convective clouds, which is significantly tuned down from 0.007 in E3SMv1 to a nominal value of 0.002 in E3SMv2. This increases cloud condensate amount detrained from deep convections and increases overall cloudiness. The impact of MG2-tuning and the new convective trigger on total cloud fraction is relatively small compared to the tuning made





in CLUBB and ZM. The MG2 tuning mainly influences high latitudes, particularly along the Antarctic coastline while the new
trigger produces more clouds over most of the regions except for the tropical western Pacific and the SO between 30°S and
60°S where a reduction of clouds is noticeable.

The impact on total cloud fraction with different cloud optical properties is further examined in Figure 8. Over 60S – 60N,
consistent with Figure 7, the CLUBB tuning has led to an increase of clouds regardless of their optical properties, which results
in a better simulation of optically thick clouds ($\tau > 9.4$) and a worse simulation of optically thin clouds ($\tau < = 9.4$). The ZM
tuning response is to increase clouds with overall minor impact on optically thin clouds ($\tau < 3.6$) and considerably improved
clouds with cloud optical depth between 3.6 and 23.  The MG tuning mainly affect simulation of intermediate clouds ($3.6 < \tau$
$< 23$) where the overestimation of clouds is reduced. The impact of the new trigger on clouds is more clearly demonstrated
when examining clouds with different optical properties. It acts to reduce optically thin clouds and increase optically
intermediate and thick clouds. This is consistent with Xie et al. (2019), which showed that the new trigger helped suppress
light precipitation and enhance intermediate and heavy precipitation.

Similar impacts of these changes are found over the SO. The tunings made in CLUBB help make optically intermediate and
thick clouds ($\tau > 9.4$) closer to the observations and the MG tuning dramatically improves optically intermediate clouds ($9.4$
$< \tau < 23$). The new trigger acts to reduce optically thin clouds and increase optically intermediate and thick clouds like over
60S to 60N. The exception is for the ZM tuning, which has minor impact over the SO.

## 4.2 CALIPSO vertical cloud structure

Figures 9-11 display the impact of these changes on the simulation of high, middle, and low clouds, respectively. For high
clouds (Figure 9), the ZM tuning has the largest impact, which considerably increases high clouds globally, especially in the
tropics. The new trigger leads to a considerable reduction of high clouds over the western Pacific, which is consistent with the
reduction in precipitation over this region as shown in Xie et al. (2019). The CLUBB tuning typically increases high clouds
over land and reduces them over ocean. Overall, the impact of MG2 on high clouds is minor with slight increase of clouds in
the Arctic and reduction of clouds along the Antarctic coastlines.

For middle clouds (Figure 10), the CLUBB tuning leads to a slight reduction of cloud amount globally. The MG2 tuning leads
to a considerable reduction of clouds in both Arctic and Antarctic and a slight increase of clouds over the SO. The impact of
the ZM tuning and the new trigger is mainly to increase clouds in the tropics (the ZM tuning) and the SO and the Arctic regions
(the new trigger).

For low clouds (Figure 11), the CLUBB tuning plays the largest role in the E3SMv2 cloud changes. It dramatically reduces
the low clouds in the tropical and subtropical regions except for the stratocumulus regime where low clouds along the
subtropical coasts largely increase. The tuning made in MG2 and ZM as well as the use of the new trigger all help offset the
reduction of clouds made by the CLUBB tuning over the tropical and midlatitude regions. The exception is over the SO where
the new trigger acts to reduce low clouds and MG2 tuning to substantially increase low clouds around the Antarctica.



The analysis of the vertical cloud structure clearly indicates compensation effect of cloud changes in the vertical that has impact on the total cloud fraction shown in Figure 7. For instance, the opposite sign of changes in middle and low clouds over the SO with the new trigger leads to overall small changes over that region in Figure 7.

### 4.3 MODIS high clouds

MODIS simulator output further provides information on the impact of these updates on high clouds with different optical properties (Figures 12 - 14). All the changes have minor impact on optically thin high clouds (Figure 12). The CLUBB and MG2 tuning and the new trigger also have minor impact on the optically intermediate and thick clouds (Figures 13 and 14), except for the notable reduction over the Western Pacific with the new trigger, which is clearly responsible for the reduction of total high clouds over this region seen in Figure 9d, and the slight reduction over the SO and N. H. storm tracks with MG2

tuning. It is interesting to see that CLUBB tuning shows slight increase in optically thin and intermediate land high clouds, consistent with that seen with CALIPSO analysis. The major impact on high clouds is from the ZM tuning, which largely increases the optically intermediate and thick clouds. This indicates that the changes in high clouds we see in Figure 9 with CALIPSO are mainly in intermediate and thick clouds.

### 4.4 MISR low clouds

Figures 15-17 display differences in low clouds from MISR simulator between sensitivity tests and the E3SMv2 for optically thin, intermediate, and thick clouds, respectively. For optically thin clouds ($0.3 < \tau < 3.6$), the largest changes are seen from the use of the new trigger, which largely reduce cloud amounts near the Antarctic coast and in the high latitude in the N. Hemisphere indicating that the reduction of optically thin clouds shown in Figure 8 from the new trigger are mainly from low clouds.  In contrast, the MG2 tuning notably increases low clouds in the high latitudes and over the SO. The impact of the

CLUBB and ZM tuning on optically thin clouds are minor.

For optically intermediate clouds ($3.6 < \tau < 23$) shown in Figure 16, the changes are dominated by the CLUBB tuning, which dramatically reduces the low clouds in the tropical and subtropical regions. This suggests the reduction of low clouds seen in Figure 11a is from the optically intermediate clouds.

It is interesting to see that the new trigger largely increases optically thick low clouds near the Antarctic coast and in the

northern high latitudes (Figure 17), whereas changes of opposite sign are seen in its produced optically thin clouds as shown in Figure 15, leading to a much smaller overall changes over these areas in Fig 11d. The tuning of MG2 consistently increases optically thick clouds over the SO and the northern high latitudes. The impact of the CLUBB tuning and ZM on the optically thick clouds is minor.

### 4.5 SWCRE and LWCRE



The impacts of these model changes on SWCRE and LWCRE are shown in Figures 18 and 19, respectively. For SWCRE, consistent with the large reduction of low clouds, the CLUBB tuning largely reduces SWCRE over the tropical and subtropical oceans. In contrast, the SWCRE in the tropical deep convection regions becomes much stronger mainly due to the increase of optically intermediate and thick high clouds due to the ZM tuning. The reduction of high clouds over the Western Pacific with the new trigger leads to reduction of SWCRE. Overall, the impact of the MG2 tuning on SWCRE is minor.

For LWCRE, only the ZM tuning has noticeable impact, leading to an increase in LWCRE due to the increase of high clouds. The minor impact of the CLUBB tuning in LWCRE indicates the changes in low clouds have minor impact on LWCRE.

## 5 Summary

We performed systematic evaluation of clouds simulated in the newly developed Energy Exascale Earth System Model (E3SM) version 2 (E3SMv2) with satellite observations by utilizing the satellite simulator package (COSP) to mitigate
sampling and algorithmic differences between modeled and observed clouds. Multiple cloud observations measured by various instruments were used to address potential data uncertainty and instrument and retrieval limitations. Our focus is to document E3SMv2 performance on clouds and understand what updates in E3SMv2 have caused the changes in clouds from E3SMv1 to E3SMv2. For the second purpose, results from 4 sensitivity tests conducted in Qin et al. (2023) were used to isolate the impact of tuning made in CLUBB, MG2 and ZM, as well as the use of the new dCAPE_ULL trigger in E3SMv2 on clouds.

In general, E3SMv2 shows a similar error pattern in clouds as exhibited in its previous version. One robust improvement is seen in the stratocumulus regime where stratocumulus clouds along the subtropical west coast of continents in both Hemispheres are largely increased. This is true by comparing all the satellite datasets used in this study and consistent with a much stronger (better) SWCRE over these stratocumulus regions. The MISR data further indicates that the improvement is mainly from optically intermediate low clouds ($3.6 < \tau < 23$) along the subtropical coasts. Note that the lack of stratocumulus
clouds is one of the most outstanding problems in E3SMv1. This improvement represents a big achievement made in E3SMv2 (Golaz et al. 2022). Relative to CALIPSO, however, E3SMv2 shows a larger negative bias in total low clouds than E3SMv1 in other regions in tropical and subtropical oceans. A comparison with MISR suggests that the smaller biases in E3SMv1 partially result from error compensation in its simulated low clouds with different optical properties, for which E3SMv1 shows an underestimation of optically thin low clouds (similar to E3SMv2) while it largely overestimates optically intermediate low
clouds in these regions. Relative to the large changes in low clouds, E3SMv1 and E3SMv2 show very similar simulation of CALIPSO high clouds although some regional changes are seen in their simulated optical properties as demonstrated in comparison of MODIS.

The sensitivity tests indicate that the improved stratocumulus along the coast is primarily from the re-tuning of parameters related to CLUBB. Other changes only have minor contributions. However, the CLUBB tuning also resulted in a reduction of
low clouds, due to the reduction of optically intermediate clouds, in the tropical and subtropical regions. This change reduced the overpredicted cumulus clouds in the subtropical cumulus regions but exaggerated the issue with underpredicted cloud in



other regions in tropical and subtropical oceans. This led to an overall slight degradation of cloud simulation in E3SMv2 compared to E3SMv1 although the better performance in the latter is partially due to error compensation in clouds with different optical properties as discussed earlier.

The sensitivity tests indicate that the impact of the MG tuning on clouds is mainly in the high latitudes over both Hemispheres where it increased low clouds and decreased middle clouds. Over the SO, its overall effect is minor. However, a close look at the cloud optical properties over the SO shows a significant improvement in optically intermediate clouds compared to MODIS. Overall, its impact on clouds is smaller than the other updates examined in this study. The ZM tuning primarily increased optically intermediate and thick high clouds over the tropical deep convection regions. The increase in the high

clouds partially offset the decrease of low clouds by the CLUBB tuning in the total cloud fraction and had a positive impact on simulation of both SWCRE and LWCRE in E3SMv2 over the tropical deep convection regions. Similar to the MG2 tuning, the impact of the new trigger on clouds is also smaller than the tuning made in CLUBB and ZM. The most noticeable change using the new trigger is the large increase in optically thick low clouds near the Antarctic coast and in the northern high latitudes, whereas changes of opposite sign are seen in its produced optically thin clouds, leading to a much smaller overall

changes over these areas in Figure 12d.

This study offered additional insights about clouds simulated in E3SMv2 by utilizing multiple data sets and the COSP diagnostic tool and through sensitivity tests. The improved understanding will benefit future E3SM developments and application of E3SM in various science applications.

**Code and data availability**

The E3SM source code is available on GitHub: https://github.com/E3SM- Project/E3SM (E3SM Project, 2023) under the 3-Clause BSD Open Source license (https://e3sm.org/resources/policies/open-source-license/). The simulations of version 1 and version 2 are reproduced using maintenance branches maint-1.0 at https://github.com/E3SM-Project/E3SM/tree/maint-1.0 (last access: April 2023; Rasch et al., 2019) and maint-2.0 at https://github.com/E3SM-Project/E3SM/tree/maint-2.0 (last access: April 2023; Golaz et al., 2022), respectively. These model codes and the simulator output from the model simulations

are archived at https://doi.org/10.5281/zenodo.8021851. The simulator output is also accessible at https://portal.nersc.gov/project/e3sm/yuying/E3SMv2/ModelOutput.

The original cloud observations for model evaluation and CERES are available at https://climserv.ipsl.polytechnique.fr/cfmip-obs and https://asdc.larc.nasa.gov/project/CERES/CERES_EBAF_Edition4.1, respectively. The observational climo data are also available at https://portal.nersc.gov/project/e3sm/yuying/E3SMv2/observation.



## Author contribution

YZ and SX initiated this study and led the analysis and manuscript writing. YQ performed the sensitivity tests. QT and XZ performed the E3SM v1 and v2 historical simulations. YZ coordinated the manuscript writing with input from all the co-authors.

## Competing interest

Po-Lun Ma is a Topical Editor of Geoscientific Model Development. Other authors declare that they have no conflict of interest.

## Acknowledgement

This research was primarily supported as part of the Energy Exascale Earth System Model (E3SM) project and partially supported by the PCMDI cloud feedback project of the Regional and Global Model Analysis (RGMA) Program, funded by the U.S. Department of Energy, Office of Science, Office of Biological and Environmental Research. The authors thank all E3SM team members for their efforts in developing and supporting the E3SM model version 2. Work at LLNL was performed under the auspices of the US DOE by Lawrence Livermore National Laboratory under contract No. DE-AC52-07NA27344. The Pacific Northwest National Laboratory (PNNL) is operated for DOE by Battelle Memorial Institute under contract DEAC06-76RLO 1830. This research used high-performance computing resources of the National Energy Research Scientific Computing Center, a DOE Office of Science User Facility supported by the Office of Science of the U.S. Department of Energy under Contract No. DE-AC02-05CH11231.

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



**Table 1.** Satellite data used in the study


|  | Quantity | Time Period | Data Source | Reference |
|---|---|---|---|---|
| **ISCCP[#]** | Cloud cover - joint histogram | 1983-2017 | ISCCP H | Rossow et al. 2016 Pincus et al. 2012 Zhang et al. 2012 |
| **MODIS[#]** | Cloud cover - joint histogram | 2002-2016 | Collection 5.1 | Pincus et al. 2012 |
| **MISR[#]** | Cloud cover -joint histogram | 2000-2020 | CTH-OD | Marchand et al. 2010 |
| **CALIPSO[#]** | High, middle, and low cloud fraction | 2006-2020 | GOCCP v3.1.2 | Chepfer et al. 2010 |
| **CERES** | TOA shortwave and longwave cloud radiative effect | 2000-2021 | CERES_EBAF_Ed4.1 | Loeb et al. 2018 |

#: The joint histogram of cloud cover (as a function of cloud-top pressure (*ctp*) or cloud-top height (*cth*) and optical thickness (*τ*)) and CALIPSO cloud data are from the CFMIP GCM Simulator-Oriented cloud products developed and can be downloaded from http://climserv.ipsl.polytechnique.fr/cfmip-obs/.


**Table 2. Description of simulations**

| Name | Description | Purpose |
|---|---|---|
| v1_coupled | The last 30-year (1985-2014) data from the 150-year historical simulations with E3SMv1 | E3SMv1 baseline |
| v2_coupled | The last 30-year (1985-2014) data from the 150-year historical simulations with E3SMv2 | E3SMv2 baseline |
| v2 | EAM v2 configuration: 6 yr control – AMIP | V2 baseline for sensitivity tests |
| Trigonly | V2 with ZM trigger reverted to v1, i.e., turn off the dCAPE_ULL trigger | Test the impact of the new trigger |
| clubbonly | V2 with CLUBB related parameters reverted to v1 | Test the impact of CLUBB tuning |
| MGonly | V2 with MG2 related parameters reverted to v1 (Bergeron factor, minimum CDNC, accretion factor, autoconversion factor, …) | Test the impact of MG tuning |
| ZMonly | V2 with ZM related parameters reverted to v1, dCAPE_ULL trigger retained | Test the impact of ZM tuning |



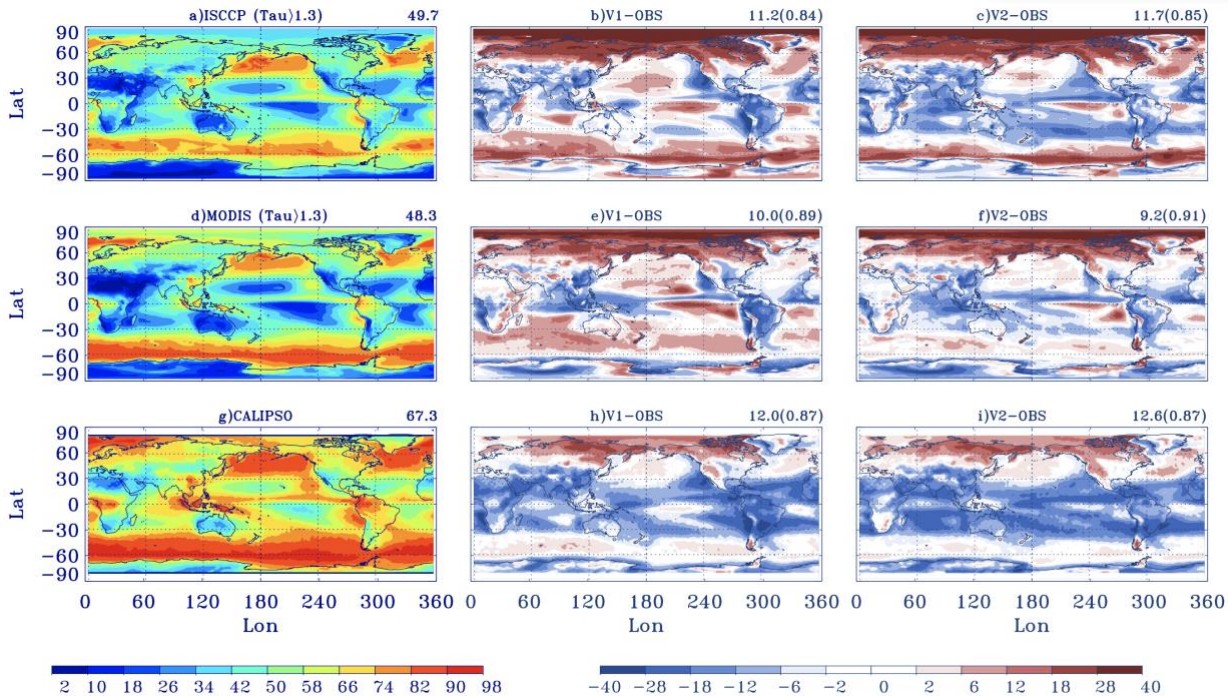

**Figure 1.** Annual mean total cloud fraction from a) ISCCP (Tau>1.3), d) MODIS (Tau>1.3), and g) CALIPSO, the differences between E3SMv1 and observations b), e), and h), and the differences between E3SMv2 and observations c), f), and i). The number after each panel name is the global annual mean of cloud fraction for observations or the global annual mean RMSE and correlation (in parentheses) for cloud fraction differences.





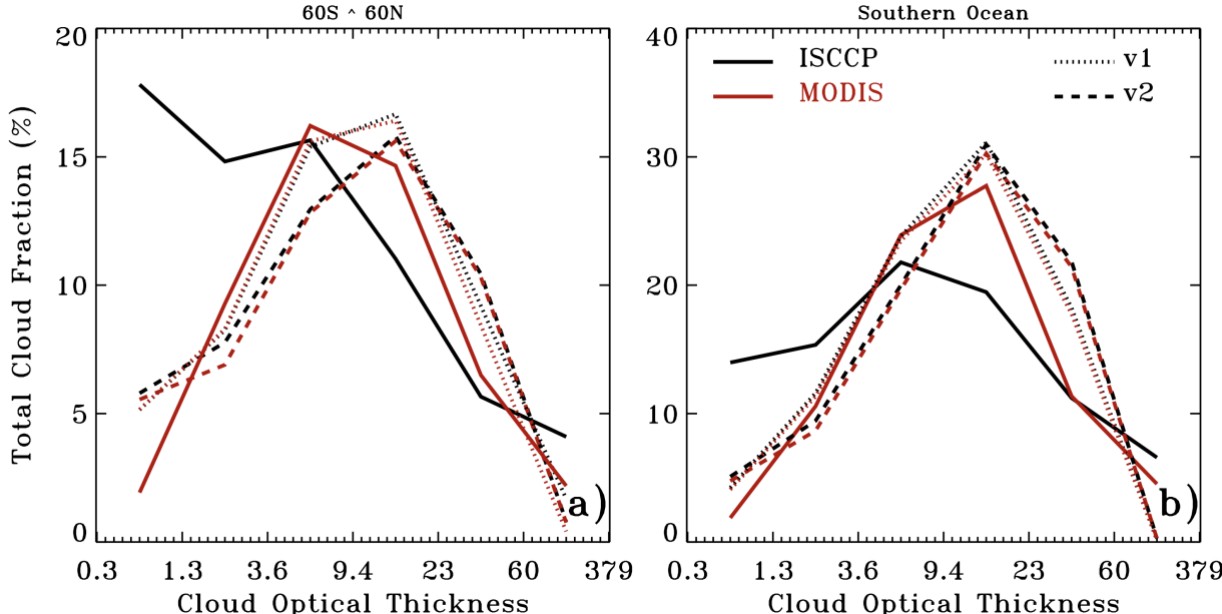


**Figure 2.** Column-integrated cloud optical depth distributions averaged a) over 60S-60N and b) over Southern Ocean (45S-60S) for ISCCP, MODIS, E3SMv1 and E3SMv2.



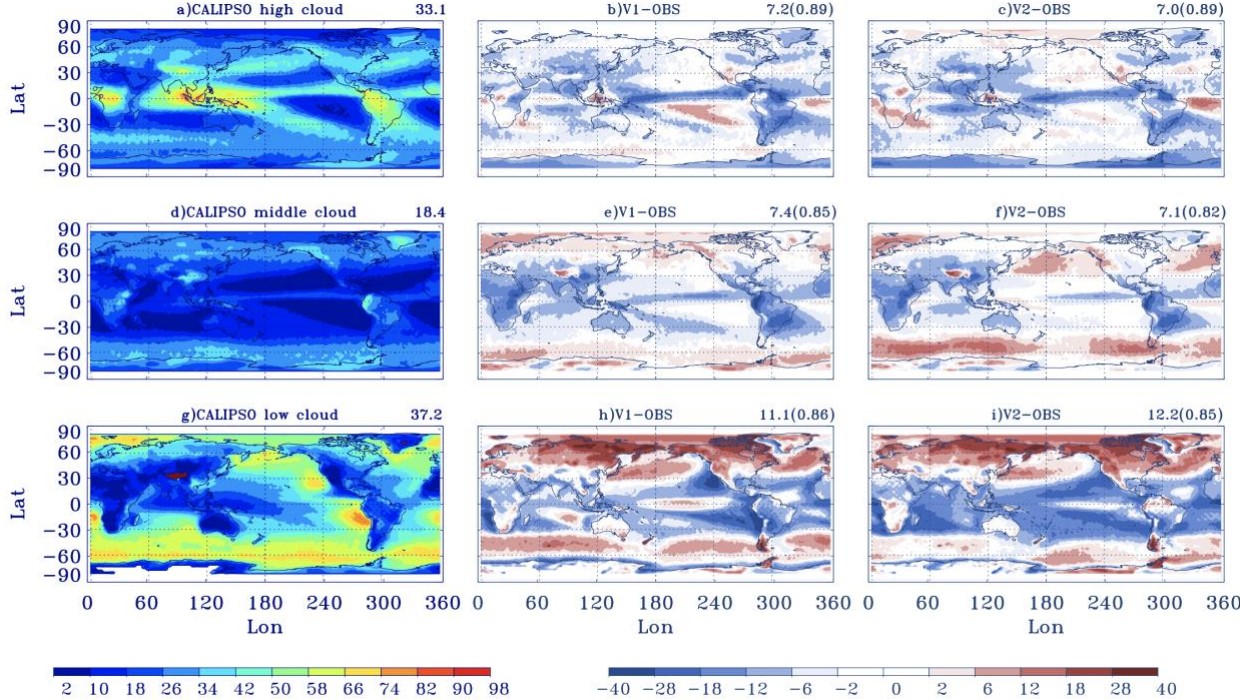


**Figure 3.** Annual mean CALIPSO-derived a) high, d) middle, g) low cloud fraction, the differences between E3SMv1 and observation for b), e), h), and the differences between E3SMv2 and observation for c), f), i). The number after each panel name is the global annual mean of cloud fraction for observations or the global annual mean RMSE and correlation (in parentheses)

for cloud fraction differences.





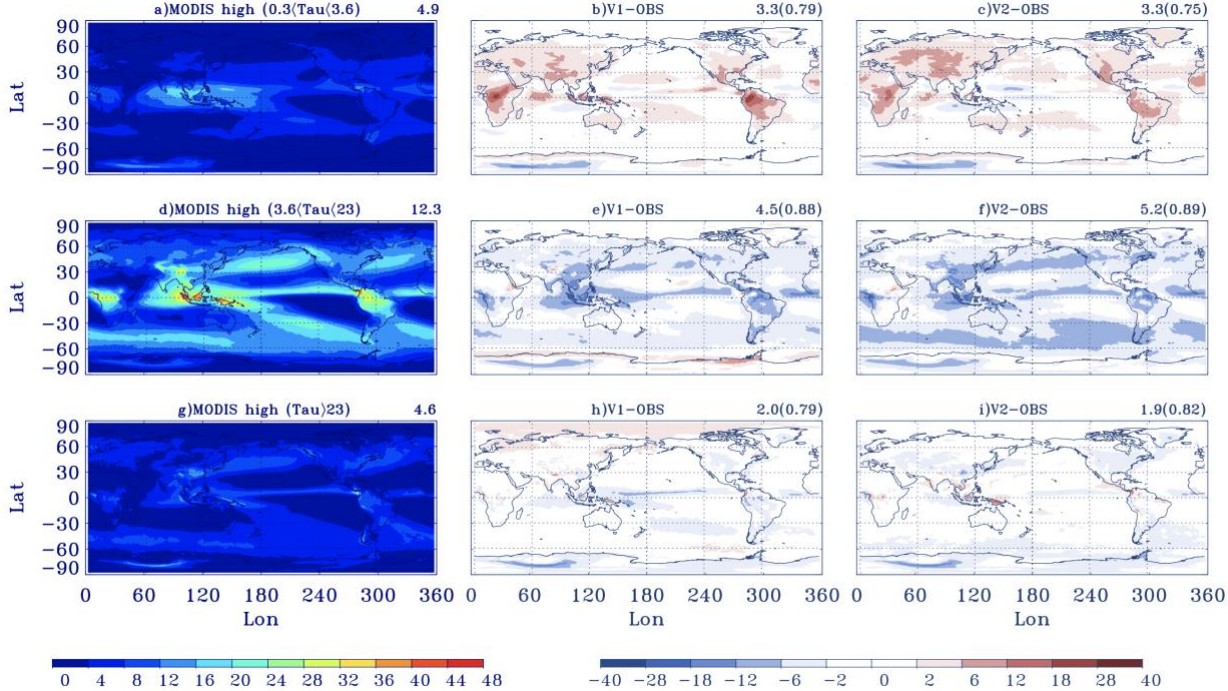

**Figure 4.** Annual mean MODIS high-topped thin (0.3<Tau<3.6), intermediate (3.6<Tau<23), and thick (Tau>23) clouds. a) & d) & g) are MODIS observations, b) & e) & h) are the differences between E3SMv1 and observation, and c) & f) & i) are the differences between E3SMv2 and observation. The number after each panel name is the global annual mean of cloud fraction for observations or the global annual mean RMSE and correlation (in parentheses) for cloud fraction differences.





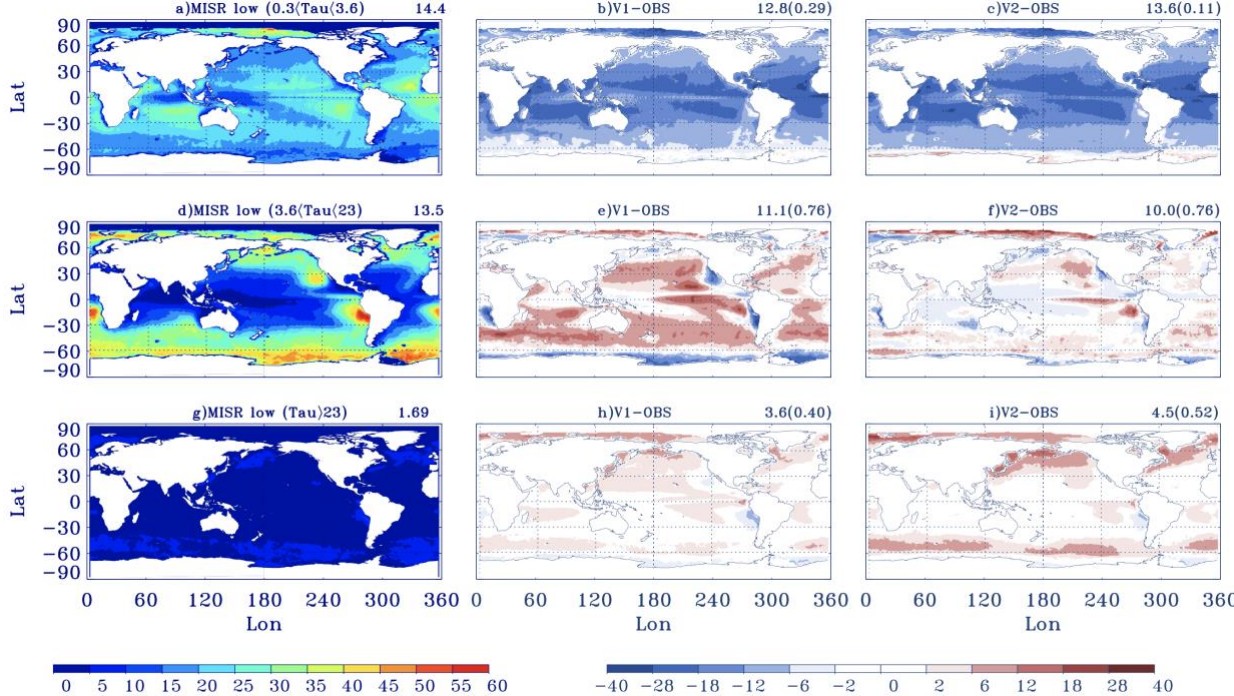

**Figure 5.** Annual mean MISR low-topped optically thin (0.3<Tau<3.6, intermediate thickness (3.6<Tau<23) and thick (Tau
> 23) clouds. a) & d) & h) are MISR observations, b) & e) & h) are the differences between E3SMv1 and observation, and c)
& f) & i) are the differences between E3SMv2 and observation. The number after each panel name is the global annual mean
of cloud fraction for observations or the global annual mean RMSE and correlation (in parentheses) for cloud fraction
differences.





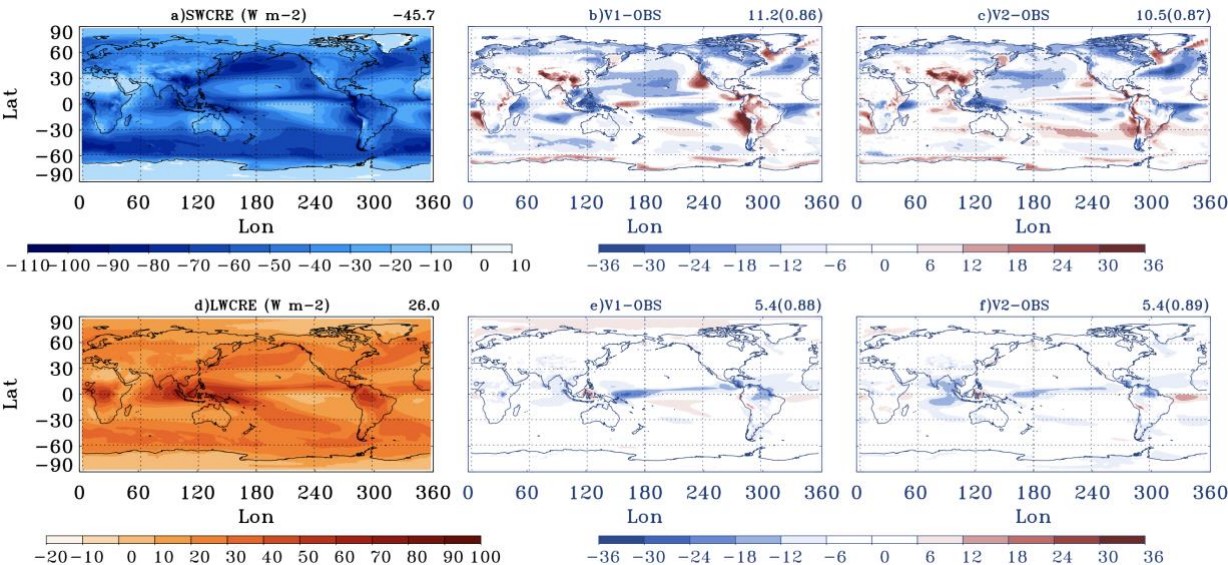

**Figure 6.** Annual mean a) SWCRE and d) LWCRE for CERES, and b) and e) differences between E3SMv1 and CERES, and c) and f) between E3SMv2 and CERES. The number after each panel name is the global annual mean of cloud fraction for observations or the global annual mean RMSE and correlation (in parentheses) for cloud fraction differences.



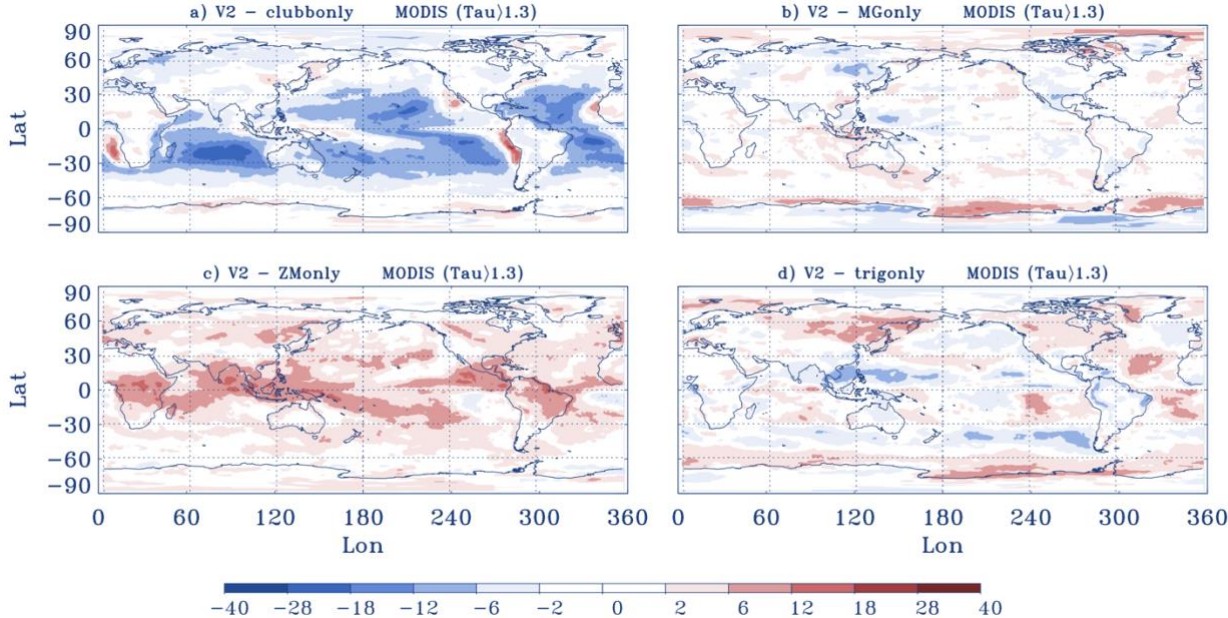

**Figure 7.** Difference in cloud fraction from MODIS simulator (Tau>1.3) between sensitivity tests and the default E3SMv2 run. a) E3SMv2 with CLUBB related parameters changed from v2 to v1; b) E3SMv2 with MG2 related parameters changed from v2 to v1; c) E3SMv2 with ZM related parameters changed from v2 to v1; and d) E3SMv2 with the dCAPE_ULL trigger turned off. All results are from 6-year AMIP-style climatology runs.



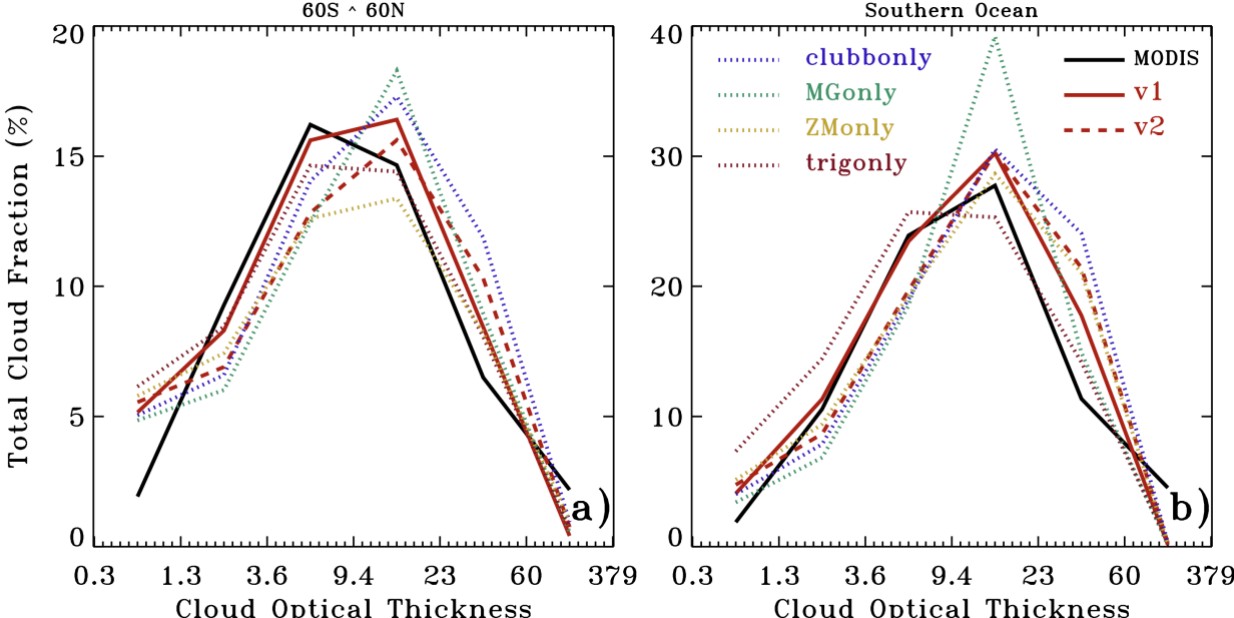

**Figure 8.** Column-integrated cloud optical depth distributions averaged a) over 60S-60N and b) over Southern Ocean (45S-60S) for MODIS, E3SMv1, E3SMv2, and sensitivity tests.



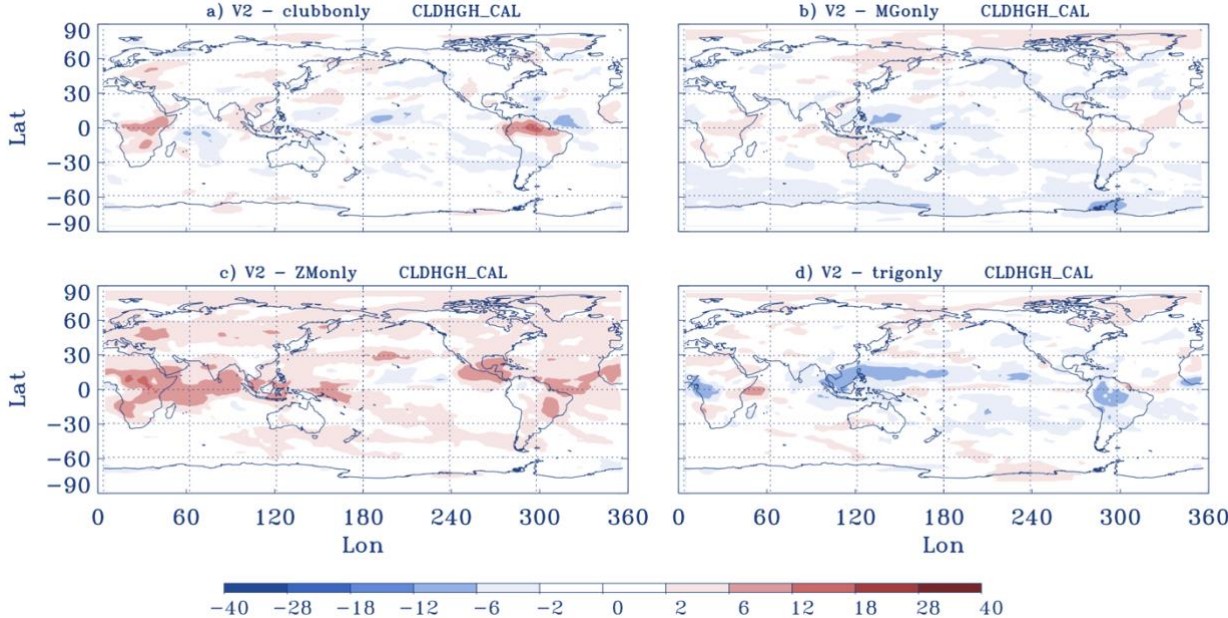

**Figure 9.** Difference in high cloud fraction from CALIPSO simulator between sensitivity tests and the default E3SMv2 run. a) E3SMv2 with CLUBB related parameters changed from v2 to v1; b) E3SMv2 with MG2 related parameters changed from v2 to v1; c) E3SMv2 with ZM related parameters changed from v2 to v1; and d) E3SMv2 with the dCAPE_ULL trigger turned off. All results are from 6-year AMIP-style climatology runs.



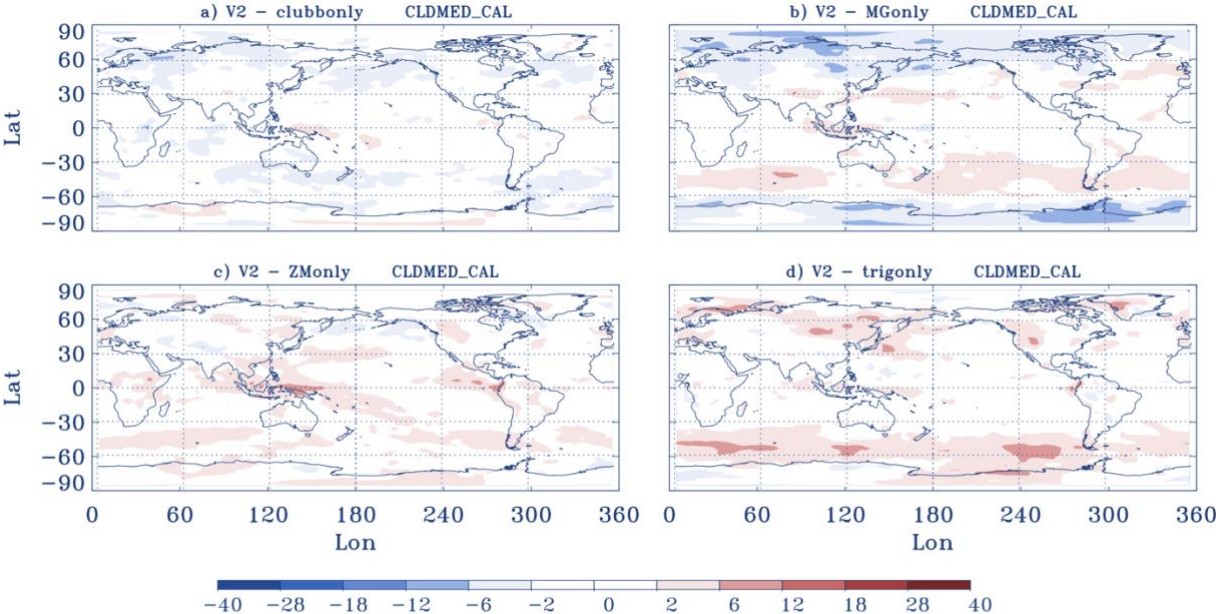


**Figure 10.** Difference in middle-level cloud fraction from CALIPSO simulator between sensitivity tests and the default E3SMv2 run. a) E3SMv2 with CLUBB related parameters changed from v2 to v1; b) E3SMv2 with MG2 related parameters changed from v2 to v1; c) E3SMv2 with ZM related parameters changed from v2 to v1; and d) E3SMv2 with the dCAPE_ULL trigger turned off. All results are from 6-year AMIP-style climatology runs.






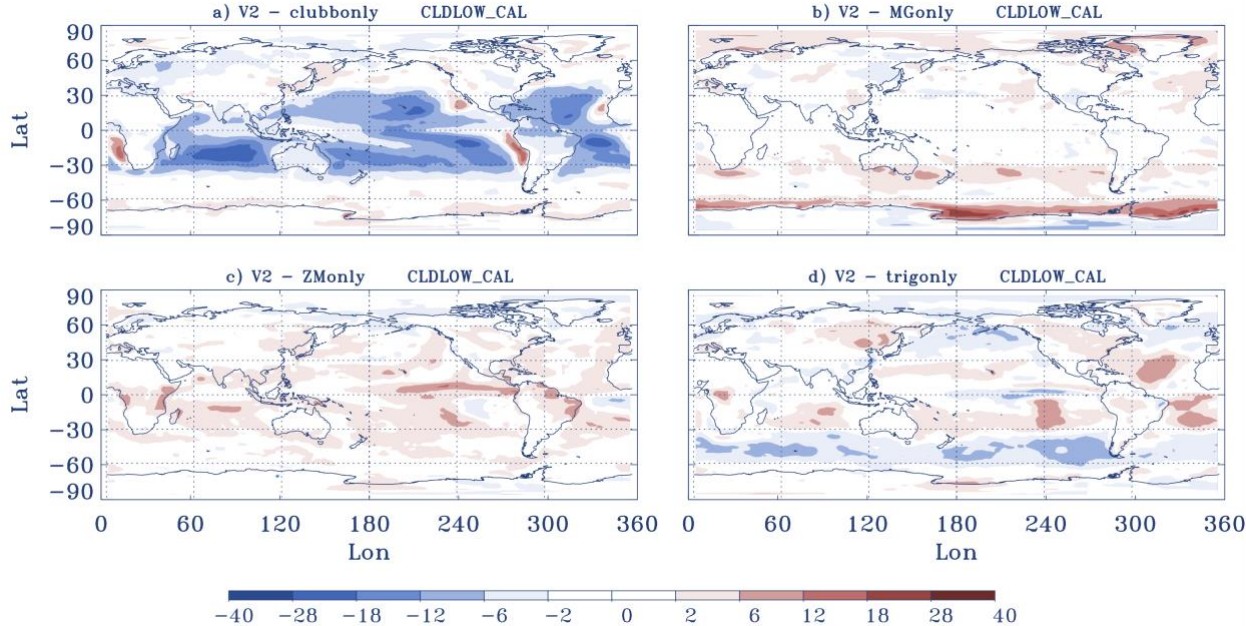

**Figure 11.** Difference in low cloud fraction from CALIPSO simulator between sensitivity tests and the default E3SMv2 run. a) E3SMv2 with CLUBB related parameters changed from v2 to v1; b) E3SMv2 with MG2 related parameters changed from v2 to v1; c) E3SMv2 with ZM related parameters changed from v2 to v1; and d) E3SMv2 with the dCAPE_ULL trigger turned off. All results are from 6-year AMIP-style climatology runs.







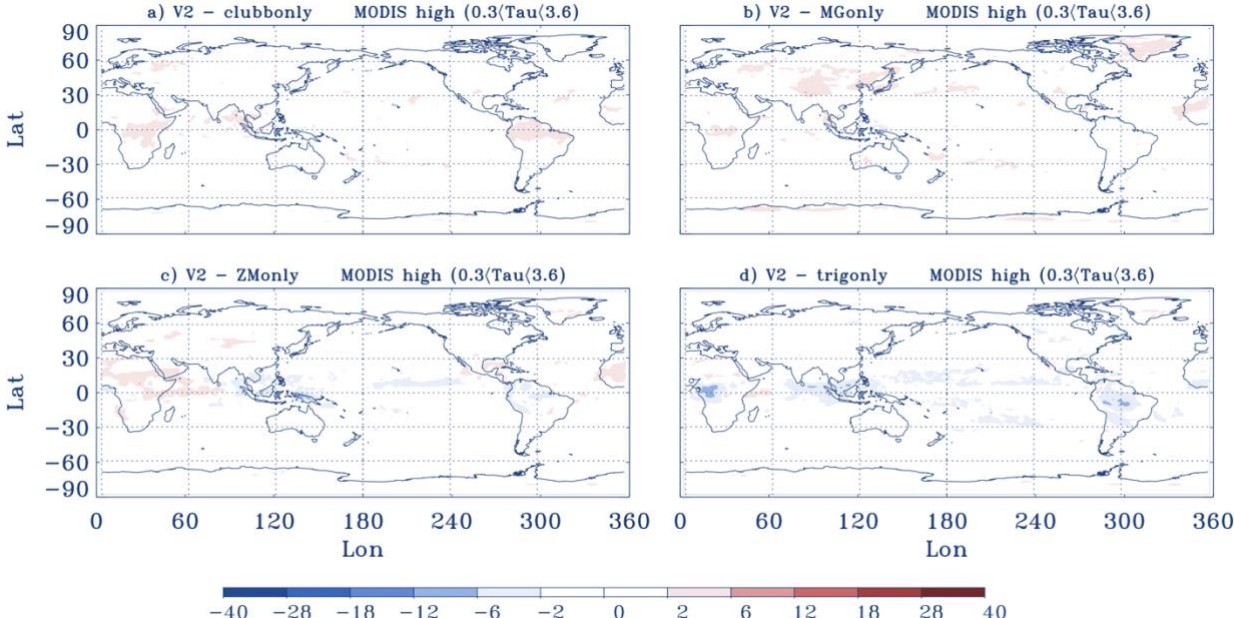

**Figure 12.** Difference in optically thin high cloud fraction from MODIS simulator (0.3 < Tau < 3.6) between sensitivity tests and the default E3SMv2 run. a) E3SMv2 with CLUBB related parameters changed from v2 to v1; b) E3SMv2 with MG2 related parameters changed from v2 to v1; c) E3SMv2 with ZM related parameters changed from v2 to v1; and d) E3SMv2 with the dCAPE_ULL trigger turned off. All results are from 6-year AMIP-style climatology runs.



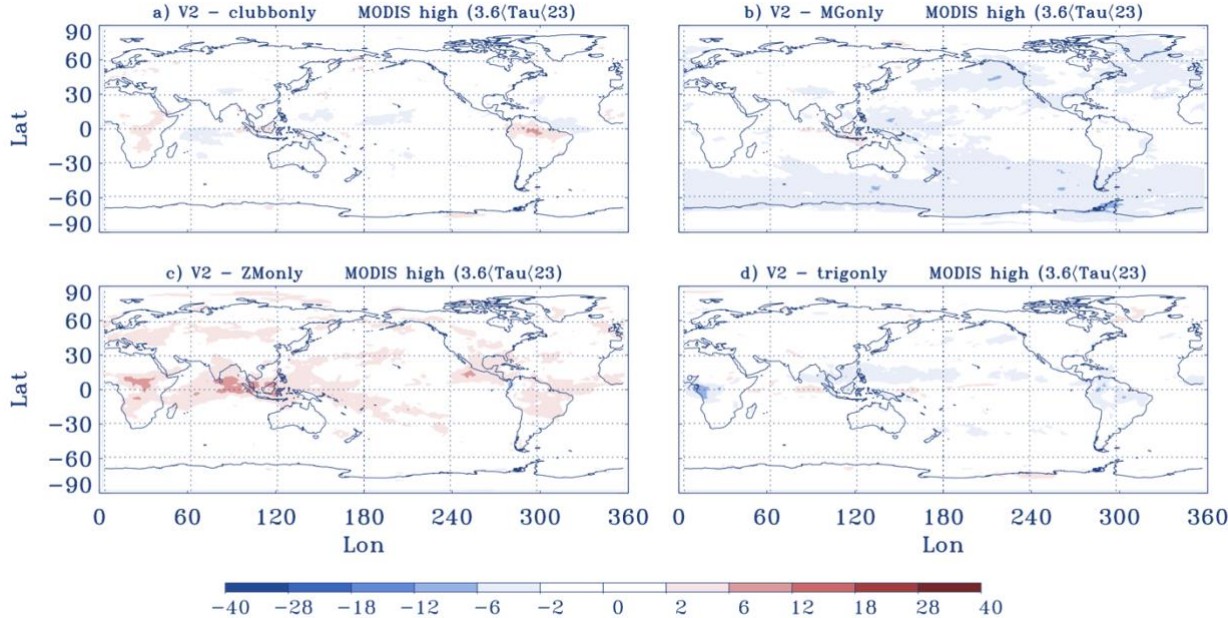

**Figure 13.** Difference in optically intermediate high cloud fraction from MODIS simulator (3.6<Tau <23) between sensitivity tests and the default E3SMv2 run. a) E3SMv2 with CLUBB related parameters changed from v2 to v1; b) E3SMv2 with MG2 related parameters changed from v2 to v1; c) E3SMv2 with ZM related parameters changed from v2 to v1; and d) E3SMv2 with the dCAPE_ULL trigger turned off. All results are from 6-year AMIP-style climatology runs.




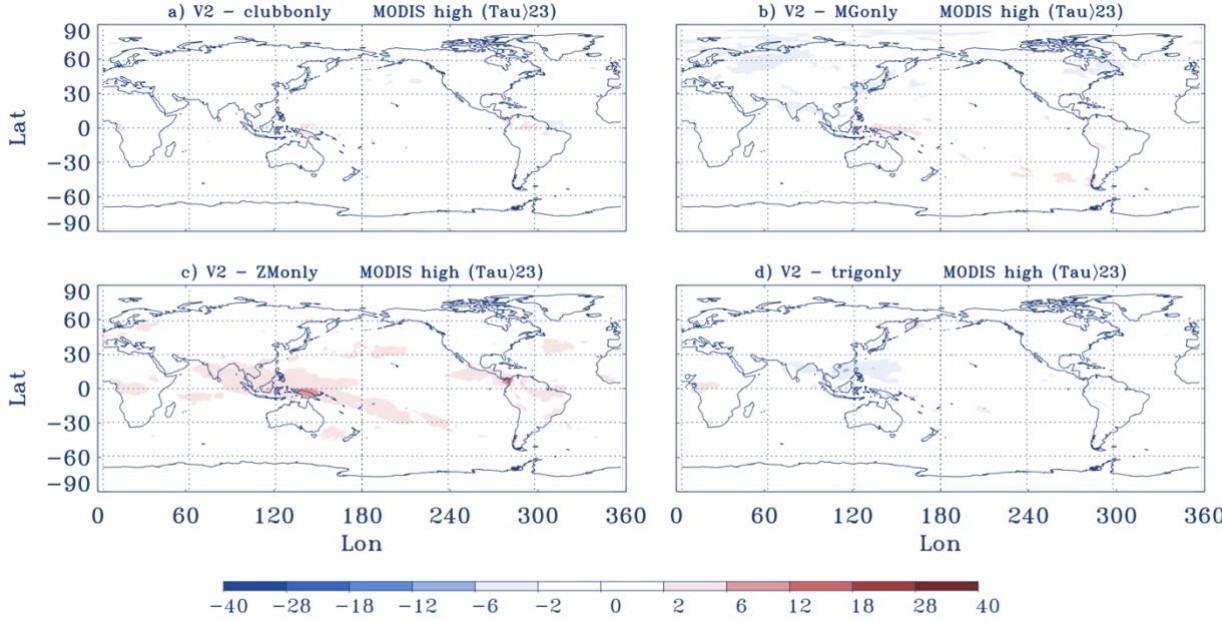

**Figure 14.** Difference in optically thick high cloud fraction from MODIS simulator (Tau >23) between sensitivity tests and the default E3SMv2 run. a) E3SMv2 with CLUBB related parameters changed from v2 to v1; b) E3SMv2 with MG2 related parameters changed from v2 to v1; c) E3SMv2 with ZM related parameters changed from v2 to v1; and d) E3SMv2 with the dCAPE_ULL trigger turned off. All results are from 6-year AMIP-style climatology runs.





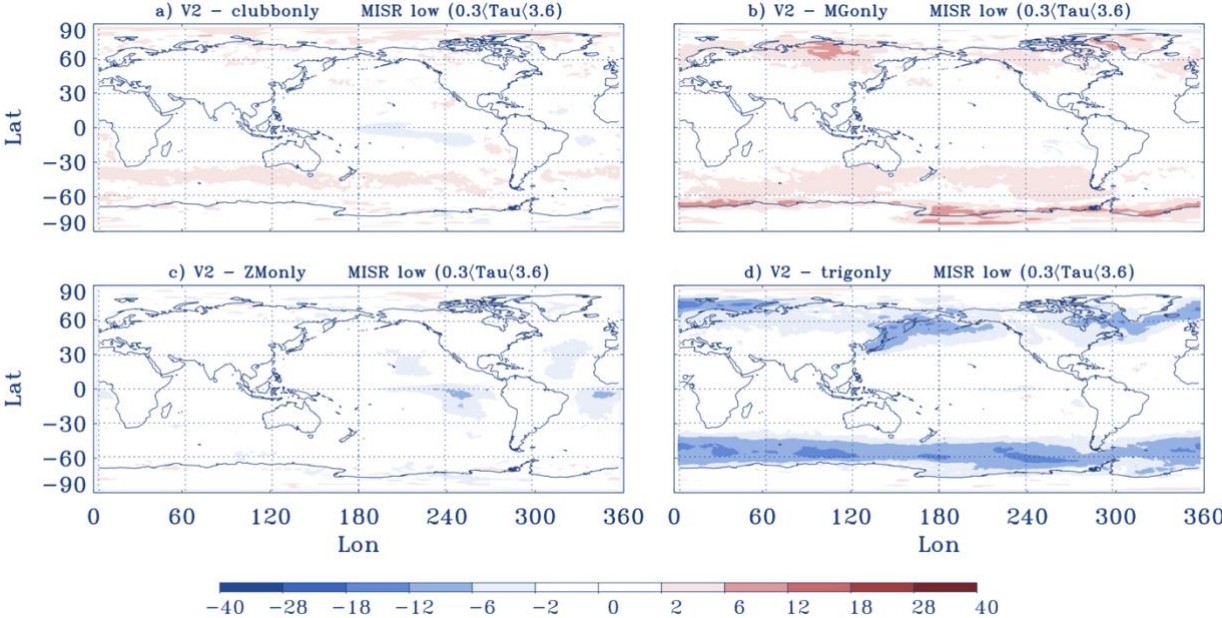

**Figure 15.** Difference in optically thin cloud fraction from MISR simulator ($0.3 < \text{Tau} < 3.6$) between sensitivity tests and the default E3SMv2 run. a) E3SMv2 with CLUBB related parameters changed from v2 to v1; b) E3SMv2 with MG2 related parameters changed from v2 to v1; c) E3SMv2 with ZM related parameters changed from v2 to v1; and d) E3SMv2 with the dCAPE_ULL trigger turned off. All results are from 6-year AMIP-style climatology runs.





625

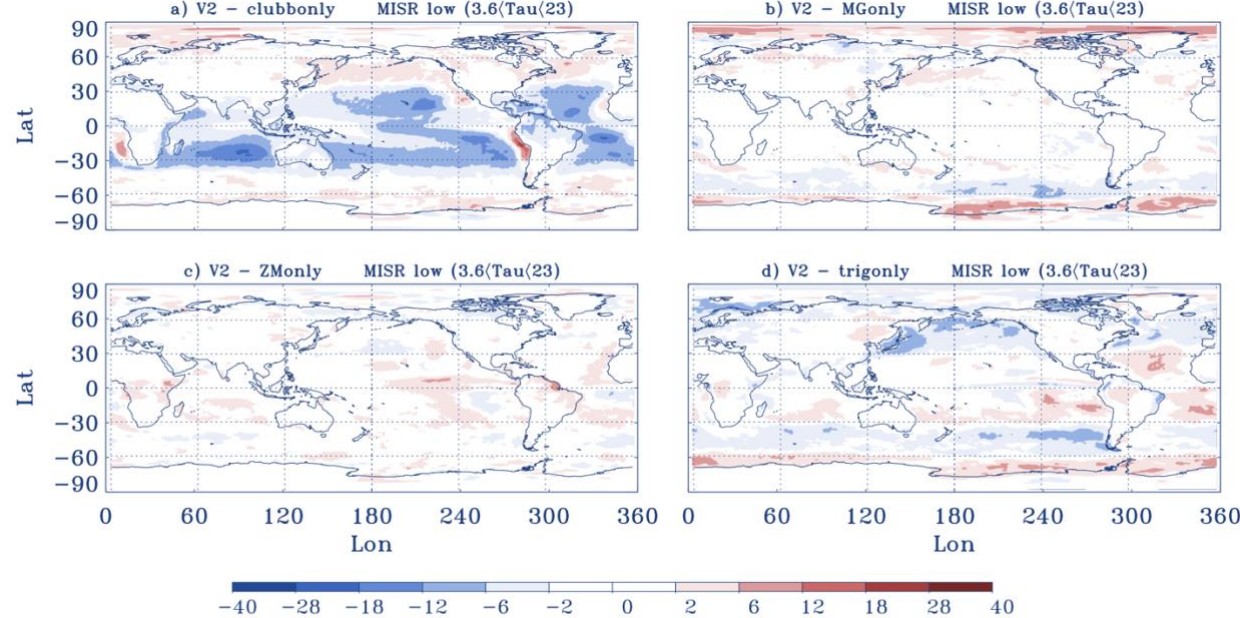

**Figure 16.** Difference in optically intermediate cloud fraction from MISR simulator (3.6 < Tau < 23) between sensitivity tests and the default E3SMv2 run. a) E3SMv2 with CLUBB related parameters changed from v2 to v1; b) E3SMv2 with MG2 related parameters changed from v2 to v1; c) E3SMv2 with ZM related parameters changed from v2 to v1; and d) E3SMv2 with the dCAPE_ULL trigger turned off. All results are from 6-year AMIP-style climatology runs.



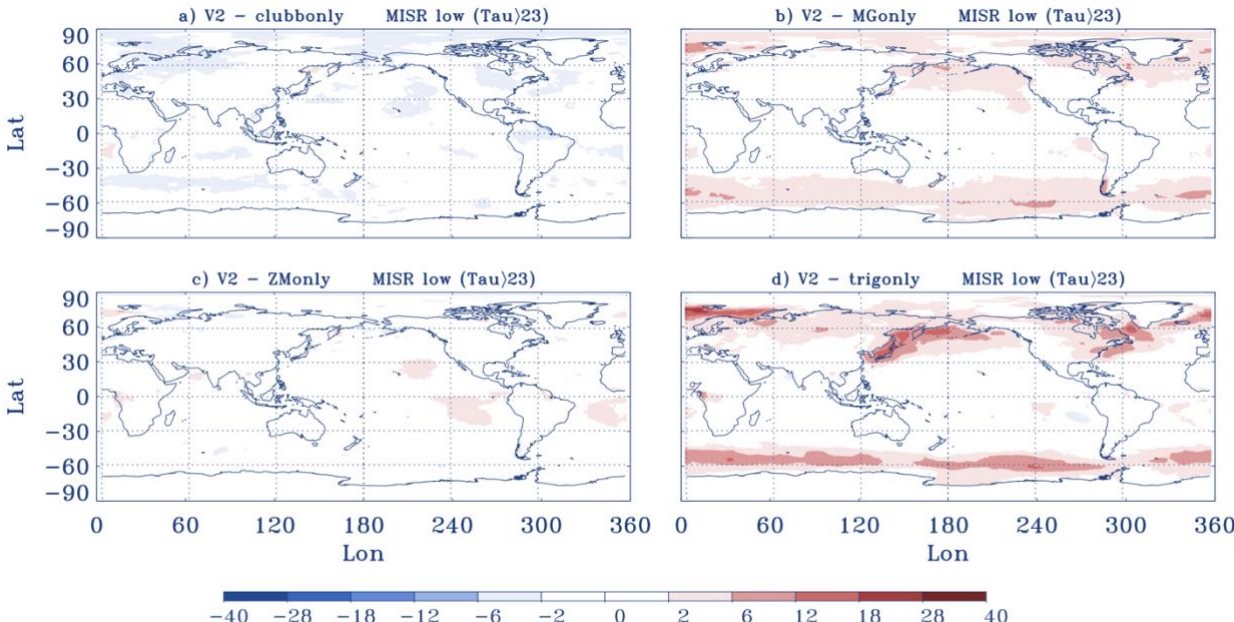

**Figure 17.** Difference in optically think cloud fraction from MISR simulator (Tau > 23) between sensitivity tests and the default E3SMv2 run. a) E3SMv2 with CLUBB related parameters changed from v2 to v1; b) E3SMv2 with MG2 related parameters changed from v2 to v1; c) E3SMv2 with ZM related parameters changed from v2 to v1; and d) E3SMv2 with the dCAPE_ULL trigger turned off. All results are from 6-year AMIP-style climatology runs.

640



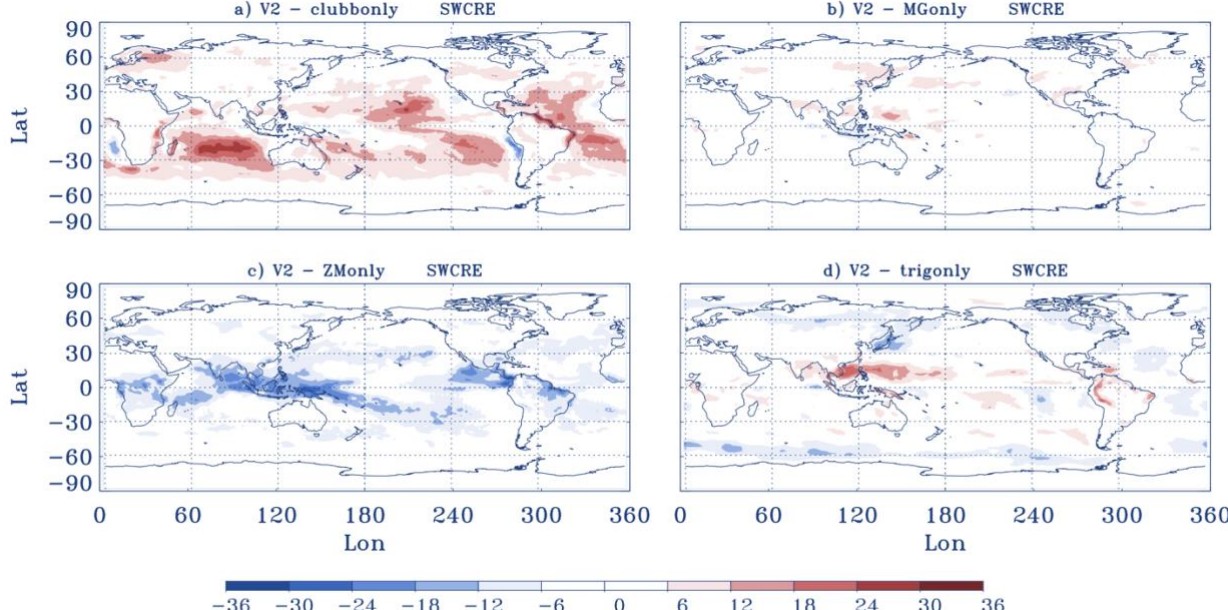

**Figure 18.** Difference in SWCRE between sensitivity tests and the default E3SMv2 run. a) E3SMv2 with CLUBB related parameters changed from v2 to v1; b) E3SMv2 with MG2 related parameters changed from v2 to v1; c) E3SMv2 with ZM related parameters changed from v2 to v1; and d) E3SMv2 with the dCAPE_ULL trigger turned off. All results are from 6-year AMIP-style climatology runs.





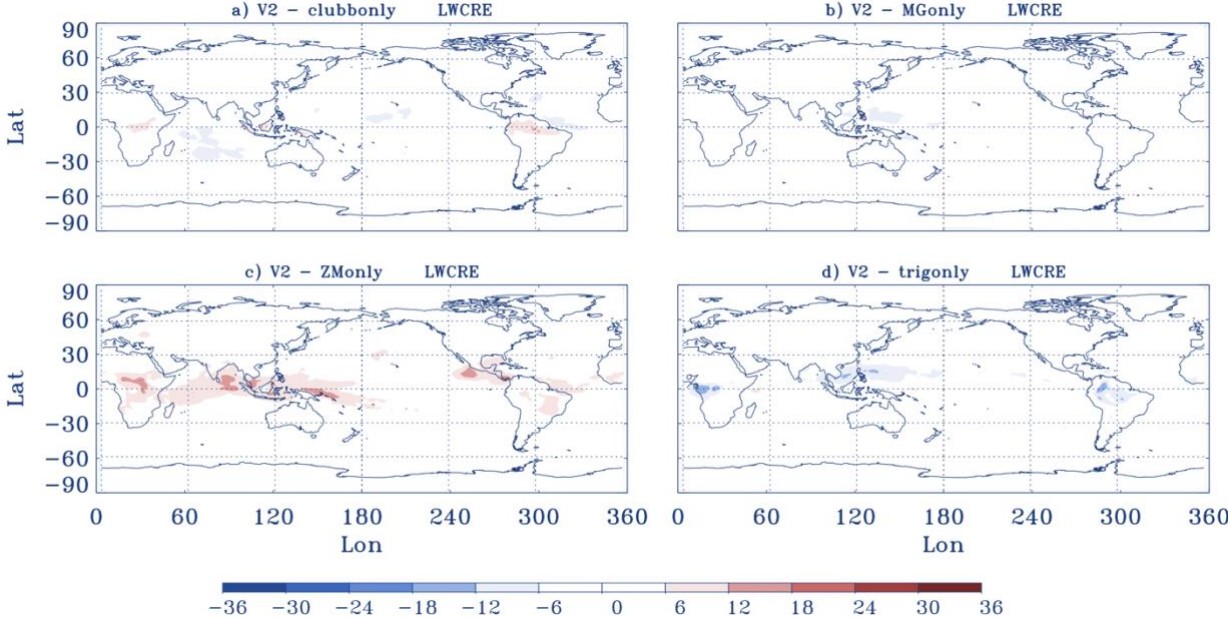

**Figure 19.** Difference in LWCRE between sensitivity tests and the default E3SMv2 run. a) E3SMv2 with CLUBB related parameters changed from v2 to v1; b) E3SMv2 with MG2 related parameters changed from v2 to v1; c) E3SMv2 with ZM related parameters changed from v2 to v1; and d) E3SMv2 with the dCAPE_ULL trigger turned off. All results are from 6-year AMIP-style climatology runs.

650