# Peer review of "Understanding Changes in Cloud Simulations from E3SM Version 1 to Version 2"

_EGUsphere, 2023_

## Author Comment (AC1)

**Understanding Changes in Cloud Simulations from E3SM Version 1 to Version 2**
Yuying Zhang et al.

Response to reviewer 1

We thank the reviewer for his/her constructive comments, which have helped further improve the clarity and quality of the manuscript. We have made changes to the original manuscript based upon the suggestions and comments from the reviewer. Our detailed responses (blue) to the reviewer's questions and comments (Italic) are listed below.

*Reviewer 1's comments*

*This study attempts to understand cloud changes induced by the E3SM version change from version 1 (E3SMv1) to version 2 (E3SMv2) with comparisons between E3SM COSP simulator results and corresponding satellite data. Also, the authors try to decompose the cause of cloud change from changes in each physics scheme (MG2, CLUBB, and ZM). This paper seems to achieve the authors' goal of showing the cloud changes of the newly released version of the model and which process influences those changes. I think this paper is valuable to be published after some revisions.*

*Main Comment*

1. *For the sensitivity tests, it will be more helpful for the reader's understanding if the authors can explain more about the parameter tuning and dCAPE_ULL trigger (object and results, briefly) even though the authors already refer to Ma et al. (2022), Golaz et al. (2022), and Qin et al. (2023).*

   **Response**: Follow the suggestion, we have provided more information about what are changed in the four sensitivity tests. See the revised Section 2.3.

2. *I think some modifications can improve Section 4. Section 4 comprises many figures and explanations for the figures, but the objective for Section 4 does not seem clear to me. I think there might be two different ways:*

   ○ *Simplify Section 4 and concentrate more on the connection between Figures 7~11 and 12~17.*

   ○ *Describe details about how the newly adapted processes in E3SMv2 derive the differences in figures 12~17.*

   **Response**: We thank the reviewer for the constructive suggestion. Following the suggestion, we largely simplified Section 4 by removing Figures 12~17 and making a closer connection between Figures 7-11 and 12-17 in the discussion. See Section 4.1 and 4.2 in the revised manuscript.

3. *Some comments for specific sentences*

   o *Line 146~148: If a reference can be added, it will be helpful.*

   **Response**: Klein et al. (2013) and Zhang et al. (2019) are added.

   o *Line 196~197: If authors explain more about the "additional information", it will be helpful.*

   **Response**: More information has been added (lines 220 to 223 in the revised manuscript):
   "For instance, the degradation of total cloud fraction simulation in E3SMv2 over the tropical and subtropical regions as shown in Figure 1 is primarily due to errors in low clouds since middle and high clouds are generally improved over these regions (Figures 3c, 3f, and 3i)."

   o *Table 1: How about adding some information about temporal and spatial resolution, which might be different from each other?*

   **Response**: The information about temporal and spatial resolution is added in Table 1.

*Technical Comment*

1. *Both τ and Tau are used in the manuscript. It seems better to select one of them to avoid confusion.*

   Tau is used in the entire manuscript.

2. *Both N.H. and N. Hemisphere are used. It seems better to select one of them to avoid confusion.*

   N. Hemisphere is used in the entire manuscript.

3. *Line 200: reginal->regional* (Changed)
4. *Line 540: cloud fraction->CRE* (Changed)

---

## Author Comment (AC3)

Understanding Changes in Cloud Simulations from E3SM Version 1 to Version 2
Yuying Zhang et al.

Response to reviewer 2

We thank the reviewer for his/her positive and constructive comments on our work. The revised manuscript has included the changes suggested by the reviewer. Our detailed responses (blue) to the reviewer's questions and comments (*Italic*) are listed below.

*Minor Comment*

1. *P.5 L.158 "considerable increase of stratocumulus cloud over the eastern ocean basins along the coasts in both hemispheres"*

*It would be helpful to readers if the authors specify more explicitly where they can find the increase in the stratocumulus cloud. I assume that they can find the increase along the western coasts of South Africa and North and South America, as stated in L.231.*

**Response**:

Done. The sentence has been revised to "A robust improvement made in E3SMv2 is the considerable increase of stratocumulus cloud over the eastern ocean basins along the coasts, such as the west coasts of South Africa and North and South America."

2. *P.9 L.258 "the CLUBB tuning has led to an increase of clouds regardless of their optical properties"*

*It appears to me that the CLUBB tuning has led to a decrease, not an increase, of clouds. This is because, in Figure 8(a), the "v2" run with the tuning shows less cloud compared with the "clubbonly" run without the tuning.*

**Response**: Thanks for pointing this out. Yes, the CLUBB tuning has led to a decrease, not an increase, of clouds. This has been corrected (P10, Line 294).

3. *P.10 L.303 "indicating that the reduction of optically thin clouds shown in Figure 8 from the new trigger are mainly from low clouds"*

*Is there any information on middle clouds that supports this statement? I can see from Figure 12(d) that high clouds do not contribute to the reduction of optically thin clouds, but I could not find information on the middle clouds.*

**Response**: To answer the reviewer's question, we examined MODIS middle clouds. As you can see from the figure below (Fig. R1), the new trigger has very minor impact on optically thin middle clouds. Its impact is mainly on increasing optically intermediate and thick middle

clouds over SO and N. Hemisphere storm tracks and the Arctic regions (not shown). This supports the statement "indicating that the reduction of optically thin clouds shown in Figure 8 from the new trigger are mainly from low clouds".

[Figure]

Figure R1. Difference in optically thin middle-level cloud fraction from MODIS simulator (0.3 < Tau < 3.6) between sensitivity tests and the default E3SMv2 run. a) E3SMv2 with CLUBB related parameters changed from v2 to v1; b) E3SMv2 with MG2 related parameters changed from v2 to v1; c) E3SMv2 with ZM related parameters changed from v2 to v1; and d) E3SMv2 with the dCAPE_ULL trigger turned off. All results are from 6-year AMIP-style climatology runs.

*Typos*

- *P.1 L.11 "165-year historical simulations". Table 2 says that they are 150-year simulations.*
  **Response**: It is 165-year historical simulations from 1850 to 2014. This has been corrected in the revised manuscript. Thanks for pointing this out.

- *P.2 L.31 "re-turning" retuning?*
  **Response**: This has been changes throughout the manuscript.

- *P.2 L.41 "To archive our goal," achieve?*
  **Response**: Fixed.

- *P.4 L.116 "165-year historical simulations": Table 2 says that they are 150-year simulations.*
  **Response**: It is 165-year historical simulations from 1850 to 2014. This has been corrected in the revised manuscript. Thanks for pointing this out.

- *P.12 L.360 "Figure 12d": Figure 11d?*
  **Response**: Fixed.

- *P.22 L.535 "a) & d) & h) are MISR observations": a) & d) & g)?*
  **Response**: Fixed.

- *P.32 L.620 "optically thin cloud fraction": optically thin low cloud fraction?*
  **Response**: Changed. The figure has been moved to Appendix following Reviewer 1's suggestion.

- *P.33 L.627 "optically intermediate cloud fraction": optically intermediate low cloud fraction?*
  **Response**: Changed. The figure has been moved to Appendix following Reviewer 1's suggestion.

- *P.34 L.633 "optically think cloud fraction": optically thick low cloud fraction?*
  **Response**: Changed. The figure has been moved to Appendix following Reviewer 1's suggestion.

---

## Author Response (AR2)

Hi Nina,

Thank you for your comments on the paper. The revised manuscript has included all the technical corrections. Our responses (blue) to your comments (*Italic*) are listed below.

*- According to GMD guidelines (https://www.geoscientific-model-development.net/submission.html) abbreviations "need to be defined in the abstract and then again at the first instance in the rest of the text." Please remove all redundant abbreviation definitions:*

*A) "SO" is defined twice within the Abstract (Line 16, Line 22) and twice in the Text (Section 3, Section 4).*

Removed "SO" definition in Line 22 and Section 4.

*B) "SWCRE" and "LWCRE" are defined redundantly in Section 3 (after already being defined in the Abstract and Section 2).*

Removed those defined in Section 3.

*C) "E3SM" and "E3SMv2" are defined redundantly in Section 5 (after already being defined in the Abstract and Section 1).*

Removed those in Section 5.

*- According to GMD policy SI units should be used and spaces must be included between number and unit. Therefore please correct "440mb", "680mb" to "440 hPa", "680 hPa" etc.*

Changed "440mb" and "680mb" to "440 hPa" and "680 hPa" in Line 210-211.

*- Use the degree sign consistently throughout the manuscript (e.g. 60° N), which is currently missing at several places (Lines 167, 168, 190, 191, 293, 305 ...).*

Changed all "60N" and "60S" to "60° N" and "60° S".

*- Please add units to Figure labels where missing (e.g., Figure 12: SWCRE ...), whereby units should be written exponentially.*

Added unit to Figure label.

*- Following GMD instructions " the abbreviation "Fig." should be used when it appears in running text and should be followed by a number unless it comes at the beginning of a sentence, e.g.: "The results are depicted in Fig. 5. Figure 9 reveals that...". "*

Changed.

*- Some abbreviations are defined in the Abstract and Text, but then barely used. For example, I can spot over 20 appearances of "stratocumulus" and only 2 appearances of "Sc" after "Sc" has been defined (if you plan to keep "Sc" I would also advise you to introduce "cumulus (Cu)" abbreviation).*

Changed "Sc" to "stratocumulus".

*- Line 31: "re-turning" should be "re-tuning".*

Changed.

Thanks,
Yuying